# Deep Learning Assisted Optimization of Metasurface for Multi-Band Compatible Infrared Stealth and Radiative Thermal Management

**DOI:** 10.3390/nano13061030

**Published:** 2023-03-13

**Authors:** Lei Wang, Jian Dong, Wenjie Zhang, Chong Zheng, Linhua Liu

**Affiliations:** 1School of Energy and Power Engineering, Shandong University, Jinan 250061, China; 2Optics and Thermal Radiation Research Center, Shandong University, Qingdao 266237, China; 3Science and Technology on Optical Radiation Laboratory, Beijing 100854, China

**Keywords:** infrared stealth, multi-band stealth, metasurface, deep learning

## Abstract

Infrared (IR) stealth plays a vital role in the modern military field. With the continuous development of detection technology, multi-band (such as near-IR laser and middle-IR) compatible IR stealth is required. Combining rigorous coupled wave analysis (RCWA) with Deep Learning (DL), we design a Ge/Ag/Ge multilayer circular-hole metasurface capable of multi-band IR stealth. It achieves low average emissivity of 0.12 and 0.17 in the two atmospheric windows (3~5 μm and 8~14 μm), while it achieves a relatively high average emissivity of 0.61 between the two atmospheric windows (5~8 μm) for the purpose of radiative thermal management. Additionally, the metasurface has a narrow-band high absorptivity of 0.88 at the near-infrared wavelength (1.54 μm) for laser guidance. For the optimized structure, we also analyze the potential physical mechanisms. The structure we optimized is geometrically simple, which may find practical applications aided with advanced nano-fabrication techniques. Also, our work is instructive for the implementation of DL in the design and optimization of multifunctional IR stealth materials.

## 1. Introduction

Infrared (IR) stealth technology, aimed at reducing the thermal signature of military targets and making them less detectable by IR detectors [1,2,3], is crucial for modern military engineering. Controlling the temperature of the target in order to reduce its IR radiative intensity is a direct but challenging method to implement [4,5]. In comparison, tailoring the spectral emissivity of the object surface is much more feasible, which is the most commonly used method for IR stealth technology [6,7]. The conventional IR stealth materials consist of metals with broad-spectrum low IR emissivity [8]. Nevertheless, contemporary detection techniques have evolved to include multi-band detection methods; in addition to the IR spectrum at the atmospheric windows, laser radar and visual detection are also implemented for improved object recognition. The drawback of traditional IR stealth materials is their high reflection, making them ineffective against laser-guided weapons. Meanwhile, the broadband low emissivity of traditional IR stealth materials is incompatible with the thermal management of the target through radiative heat transfer. As a result, there is a need for advanced multi-band IR stealth technology while keeping a channel for radiative heat transfer [9,10,11,12]. The technical requirements for multi-band compatible IR stealth are as follows: (i) low spectral emissivity in the detected band (atmosphere window) of 3~5 μm and 8~14 μm for mid-infrared (MIR) stealth; (ii) high emissivity in the undetected band of 5~8 μm for radiative cooling [13,14]; (iii) high absorptivity at 1.54 μm for reducing the near-infrared (NIR) signature commonly used by laser-guided missiles [9].

Metamaterials [15], which are artificial materials composed of subwavelength structural units, can be flexibly modulated to possess unique electromagnetic properties by designing their structures and materials, such as negative refractive index [16], stealth cloak [17], phase-abrupt electromagnetic metasurfaces [18], and holograms [19]. Metasurfaces [20], the two-dimensional counterpart of metamaterials, provide great possibilities for the modulation of IR radiation properties and have been widely studied in IR stealth technology and other particular target electromagnetic properties. Before the emergence of the metasurface, one-dimensional photonic crystals (1D-PCs) were widely investigated for IR stealth; however, the high fabrication cost and poor stability of 1D-PCs make it difficult to be widely applied [21,22,23,24,25,26]. Zhu et al. [24], combining silica aerogel and 1D-PCs, developed a high-temperature IR camouflage material with efficient thermal management. With metasurfaces, a metal-insulator-metal (MIM) structure with highly tunable selective absorption/emission has been studied for IR stealth [9,27,28,29,30,31]. Namkyu et al. [28] designed a disc-shaped MIM metasurface using Au/ZnS material to compatibly achieve IR stealth and thermal management. In addition, the phase-change material Ge_2_Sb_2_Te_5_ (GST) is often used in IR stealth technology research due to its unique optical properties [12,32,33,34]. Pan et al. [12] designed two photonic structures with the materials of Si/GST/Au, achieving camouflage that is compatible with visible, mid-infrared, and laser wavelengths. In addition to these geometrically complex metasurfaces, some simpler multilayer film models without micropatterns have also been used for IR selective emission, which is appealing for practical applications. Kim et al. [35] achieved a considerable enhancement of the atmospheric window IR emission by adding a high index lossless Ge layer to the conventional double-layer selective emitter, improving the efficiency of radiative cooling. Peng et al. [6] used a simple structure of four-layer Ge/Ag film to achieve the compatibility of IR stealth and thermal management. However, it is difficult to achieve multi-band stealth compatibility with these simple thin-film stacked structures.

The design of metasurfaces with specific radiative properties is usually based on a trial-and-error approach combined with parametric scanning [36]. In addition, some traditional optimization methods, such as Genetic Algorithms [37], Particle Swarm Optimization algorithms [38], Topological Optimization algorithms [39], etc., are also widely used. However, these optimization algorithms adopt an iterative optimization process that is very time-consuming, and the final solution frequently falls into a local optimum. These problems become much more severe in the optimization of metasurfaces with multi-dimensional degrees of freedom. Traditional optimization methods for multi-band metasurface design are difficult to balance the excellent target radiation characteristics of each band. Deep Learning (DL) [40] methods based on artificial neural networks have recently become powerful tools for complex computational and inverse design problems. In the field of nanophotonics, DL has been widely used in the direction of pattern recognition, optical imaging, and structure design [41]. In recent years, deep learning has been used for the inverse design of metasurface structures and other structural systems to obtain the desired spectral response [42]. Malkiel et al. [43] built a Geometry-predicting-network (GPN) and a Spectra-predicting-network (SPN) for an “H” shaped nanophotonic structure with eight degrees of freedom to predict and design the structure. Liu et al. [44] proposed the use of Bidirectional Neural Networks (BNN) in the design of nanophotonic structures; the main advantage of this model is the excellent solution to the so-called “one-to-many” problem. Liu et al. [45] introduced the Generative Adversarial Network (GAN) [46] into the inverse design of nanophotonic structures. The pictorial design idea of GAN broke the traditional constraints of designing only nanophotonic structures with limited degrees of freedom and greatly increased the dimensionality of the design space.

Till now, metasurfaces with multi-band IR stealth and radiative thermal management are rarely studied. Despite the advantages of DL, it has not been applied to the design and optimization of multi-band compatible IR stealth nanomaterials. Herein, we propose a Ge/Ag/Ge multilayer circular-hole metasurface, as shown in Figure 1. To realize multi-band compatible IR stealth and radiative thermal management, the Rigorous Coupled Wave Analysis (RCWA) [47] and a trained BNN model are combined to optimize the geometrical parameters of the metasurface. We note that Ge and Ag have been widely used as common IR materials for the design of IR selective emitters [6,9,24,25,30]. The optimized metasurface achieves multi-band IR stealth with low emission in the two atmospheric windows (${\bar{\varepsilon }}_{3\sim 5\mu m}$ = 0.12 and ${\bar{\varepsilon }}_{8\sim 14\mu m}$ = 0.17), narrow-band high absorption at the near-infrared laser wavelength ($\varepsilon_{1.54\mu m}$ = 0.88), and ensuring the need for radiative thermal management in the non-atmospheric window (${\bar{\varepsilon }}_{5\sim 8\mu m}$ = 0.61). We also analyze the stealth performance of the structure at large incident/emission angles and the underlying physical mechanisms. The modulation of emissivity/absorptivity is mainly for the stealth of radiation intensity; the stealth for polarized radiation is not considered in this paper.

## 2. Model and Methods

### 2.1. Structure and Data Generation

The schematic diagram of the metasurface cell is shown in Figure 1. The unit structure mainly consists of three alternating Ge/Ag/Ge films covered on an Al substrate, and each layer carries an equal diameter circular hole-shaped micropattern. *L* is the unit period, *D* is the diameter of the circular hole, *H*_1_ is the thickness of the first Ge layer, *H*_2_ is the thickness of the second Ag layer, and *H*_3_ is the thickness of the third Ge layer. The spectral absorptivity generation process of the structure is combined with a deep learning strategy. The whole workflow is shown in Figure 2, which focuses on the metasurface of a Ge/Ag/Ge multilayer circular hole. Moreover, regarding the absorption spectra bands calculated for the corresponding structure, the absorption spectra consist of 34 points which include 1.54 µm and 33 points distributed at equal intervals from 3 to 14 µm. The spectral absorptivity of the metasurface is obtained by the RCWA method. In the calculation process, we used the open source software S4 [47] as the implementation of the RCWA algorithm, and the optical constants of Ge and Ag materials are taken from the references [48,49]. Before the data generation, we verified the convergence of RCWA in terms of Fourier expansion order. We reached a balance between computation accuracy and computation time. In the validation process, we chose several sets of extreme geometric parameters to calculate the absorptivity at a specific wavelength with different expansion order settings. The expansion order was set from 10 to 1000 to check the convergence of the calculation results. Each case reached relative errors within 5% using 350 expansion orders.

In the data collection, *L* was varied from 1 to 9 µm with 2 µm intervals, *D*/*L* was varied from 0 to 0.8 with 0.2 intervals, *H*_1_ was varied from 0.05 to 1.85 μm with 0.3 μm intervals, *H*_2_ was varied from 0.005 to 0.04 μm with 0.005 μm intervals, *H*_3_ was varied from 0.05 to 1.85 μm with 0.3 μm intervals. A total of 9800 sets of data were generated by calculating using the RCWA algorithm. The data were divided into two groups, 7840 sets of data for training and 1960 sets of data for testing purposes in the deep learning process. The distribution of the 9800 sets of spectral data is shown in Figure 3, and the absorption spectra corresponding to the two structures are also given in the figure. Figure 3 is just a stacking of the spectral absorptivities of the 9800 sets of data. Firstly, the absorptivity of all the 9800 sets of data nearly covers the range from 0 to 1 at the targeted wavelengths, indicating the possibly high tunability of absorptivity of the metasurface. Secondly, it is not easy to choose a set of data that satisfies our requirement, as shown by the two randomly chosen sets of data (the blue and red lines). Thus, DL is needed to help us find the most appropriate set of data in order to achieve the goal of this work. In this work, we used the open-source software S41.1 to implement the RCWA algorithm, which is in the location of https://github.com/victorliu/S4 (accessed on: 28 February 2023). For the deep neural network part, we mainly used Python 3.8 and Pytorch 1.10.

### 2.2. Deep Learning Method

Here, we use a multilayered BNN [44] model to predict the metasurface spectral absorptivity and inversely design the geometrical parameters for the target spectral response. The reason for choosing the BNN model is that there exist instances in the data set with different geometrical parameters but having the same spectral response, i.e., the nonuniqueness of the spectral response or “one-to-many” problem. Conventional artificial neural networks (ANN) are fed with the spectral response and output the designed geometrical parameters. During the training process, ANN is hard to converge when it encounters such conflicting instances due to nonuniqueness, whereas BNN can solve this problem very well. While convolutional neural networks, or GAN, are usually used for the pixelated design of metamaterials [45,50,51], this design approach usually does not use geometrical parameters to represent the unit structure of the metasurface. The BNN consists of two fully connected neural networks (a pre-trained SPN that directly connects to a GPN), as shown in Figure 4.

During the training process, the SPN is first trained to achieve an accurate prediction of the spectral absorptivity based on the geometric input parameters (*L*, *D*, *H*_1_, *H*_2_, *H*_3_) of the metasurface structure, the number of hidden layers and the number of neurons per layer are determined during the training of the neural network by minimizing the loss function, which is defined as the mean square error (MSE) between the predicted and actual absorption spectra as follows:(1)$$\mathrm{MSE}=\frac{1}{N}\sum_{i=1}^{N}{\left(y_{i}-{\tilde{y}}_{i}\right)}^{2}$$
where *N* is the number of data points in the absorption spectrum, and here *N* is 34, *y_i_* is the actual value of the absorption spectra and ${\tilde{y}}_{i}$ is the model-predicted absorption spectra value. In the subsequent training process of the GPN, a pre-trained SPN with fixed parameters is used to evaluate the predicted absorption spectra response of the structure designed by the GPN. It is worth mentioning that the loss function of the GPN compares not the geometric parameters of design but the difference between the input and output absorption spectra, as shown in Figure 4 so that different geometric parameters lead to similar absorption spectral responses will no longer confuse the neural network and the accuracy of the prediction will be significantly improved.

## 3. Results and Discussion

### 3.1. Forward Spectra-Predicting-Network to Predict the Absorption Spectra

A fully connected SPN was designed and trained with the aim of accurately predicting the spectral absorptivity of the proposed metasurface. The SPN consists of one input layer, several hidden layers, and one output layer. The SPN with 7 hidden layers and 900 nodes per layer shows optimal performance and high accuracy after the neural network architecture optimization process, as shown in Figure 5a. After 4000 epochs of training, the loss values of the training set and test set, as shown in Figure 5b, reduce to 1.262 × 10^−7^ and 1.295 × 10^−3^, respectively. The test set MSE of 1.295 × 10^−3^ corresponds to its MAE (defined as $\mathrm{MAE}=1/N\sum_{i=1}^{N}\left(y_{i}-{\tilde{y}}_{i}\right)$) of 0.0134, and the low loss value indicates that the SPN performs well during the training process. Meanwhile, after training is completed, the specific error distribution of the test set is analyzed. Figure 5c,d compares the results between the absorptivity A’ predicted by the neural network at 1.54 μm and 14 μm and the actual value A. As shown, the SPN prediction results are in high agreement with the theoretical calculation results, indicating that the SPN predicts with good accuracy for the samples in the test set. In order to evaluate the performance of the SPN more intuitively, we use MAE as an indicator to plot its error statistical distribution, as shown in Figure 5e,f, the MAE of the majority of samples is below 0.08, samples with MAE more than 0.1 only accounts for 1.22% of all tested samples. We define the prediction accuracy for the SPN as:(2)$$\mathrm{Accuracy}=1-\mathrm{MAE}=1-\frac{1}{n}\sum_{i=1}^{n}\left(y_{i}-{\tilde{y}}_{i}\right)$$

If MAE > 0.1 represents accurate prediction, then the SPN achieves a prediction accuracy of 98.78%.

In order to further demonstrate the prediction performance of the SPN more intuitively, four samples were randomly selected from the test set for comparison, as shown in Figure 6a–d. It can be seen that the two sets of data are well-fitted, which also demonstrates the superior performance of the neural network.

### 3.2. Inverse Geometry-Predicting-Network to Design the Geometric Parameters

Compared with the forward prediction of the absorption spectra, the inverse structure design is much more complicated because different geometrical parameters of the metasurface may cause the same spectra response, i.e., the “one-to-many” problem. For example, for the Ge/Ag/Ge multilayer circular-hole structure, when the circular hole diameter D is equal to 0, the absorption spectra of the structure have the same set of data regardless of the period L, which makes the inverse structure design problem more difficult. The BNN with tandem architecture is an effective way to solve the “one-to-many” problem, as shown in Figure 4. This tandem architecture is essentially a structure formed by cascading an inverse-design ANN (GPN) with a forward-predicting ANN (SPN). During the training and design process, the data flow passes through the GPN and SPN sequentially, just like the current in a tandem circuit. After the training of the SPN, the weights and bias parameters in the network are fixed and directly connected to the GPN, then the training of the GPN is executed.

In the training process of GPN, the desired target absorption spectra are used as input, and the intermediate output layer is the designed geometric parameters (*L*, *D*, *H*_1_, *H*_2_, *H*_3_). Then the pre-trained SPN gets the predicted absorption spectra according to the designed geometric parameters. The BNN architecture used in this work differs from the original BNN architecture proposed by Liu et al. [44] in that a penalized training strategy [52,53] is also used, i.e., the loss between the predicted and actual structures is added to the first 300 epochs. This approach enhances the robustness of the network in the first 300 generations so that the predicted structure of the network does not deviate from the basic physical rules. Therefore, in this paper, when training GPN, the back-propagation loss function in the first 300 epochs contains the MSE not only between the input and the predicted spectra but also between the actual and the designed structures. The loss is calculated as follows:(3)$$Loss=\left\{ \begin{array}{l} \mathrm{MSE}(R,R^{\prime })+k\cdot \mathrm{MSE}(D,D^{\prime })\;(\mathrm{epoch}\leq 300)\\ \mathrm{MSE}(R,R^{\prime })\;(\mathrm{epoch}>300) \end{array}\right.$$
where *R* is the actual input spectra and *R’* is the predicted spectrum of BNN; *D* is the true geometric parameter corresponding to the input spectra and *D*’ is the geometric parameter of GPN design; *k* is a coefficient that determines the magnitude of the influence of structure differences on the loss function, and *k* = 0.2 is taken.

After filtering and optimization, the architecture of the GPN is finally determined as 6 hidden layers and 1100 neurons per layer, and the architecture of the whole multilayer BNN is shown in Figure 7a. The loss value of the train-set and test-set after 2000 epochs are 1.991 × 10^−4^ and 2.115 × 10^−3^, respectively, and the loss graph during training is shown in Figure 7b. The test-set MSE of 2.115 × 10^−3^ corresponds to its MAE of 0.0144, and the lower loss value illustrates that the BNN performs well during training. Figure 7c,d shows the comparison plots between the desired/target input absorption spectra in the test set and those designed by BNN at 3 randomly selected wavelengths of 8.6 μm and 13.5 μm. As shown, a high agreement is achieved, proving the superior performance of the BNN. Figure 7e,f show the statistical analysis of the MAE of the test set samples. The majority of the samples’ MAE is below 0.1, and the samples with MAE higher than 0.1 only account for 3.37% of all samples, and the design accuracy rate reaches 96.63%.

### 3.3. On-Demand Design of BNN

After constructing and training the BNN model, our ultimate goal is to reversely design the geometric parameters of the structure for the desired spectra by the model. For the multi-band compatible IR stealth, the target absorption spectra need to have possibly higher absorptivity at the laser wavelength of 1.54 μm to meet the purpose of stealth for laser-guided missiles, while at the two atmospheric windows (3~5 μm and 8~14 μm) it needs to have possibly lower emissivity (or equivalently lower absorptivity according to Kirchhoff’s law) to meet the purpose of IR stealth. In addition, to meet the purpose of radiative cooling, higher emissivity (or equivalently absorptivity) in the non-atmospheric window band (5~8 μm) is needed. When designing the target spectra input, the absorptivity at 1.54 μm is set to 1, while the absorptivity in the band from 3 to 14 μm uses Gaussian-type target spectra data, which is obtained from Equation (4):(4)$$A(\lambda )=a+be^{\left(-\frac{{\left(\lambda -\lambda_{0}\right)}^{2}}{2\sigma^{2}}\right)}$$
where *a* is the minimum value of Gaussian spectra, *a* + *b* is the peak of the spectra, *λ*_0_ is the position of the center of the spectra, and *σ* determines the steepness of the peaks.

In this work, 4620 sets of eligible target spectra are generated based on the four coefficients of Gaussian function *a*, *b*, *λ*_0_, and *σ* in different ranges. The range of *a* was set to 0.05~0.15 with 0.01 intervals, the range of *b* was set to 0.8~0.85 with 0.01 intervals, the range of *λ*_0_ was set to 6~6.9 with 0.1 intervals, the range of *σ* was set to 0.6~1.2 with 0.1 intervals. The final generated 4620 sets of the target spectra are fed into BNN by batch input, and the designed structure parameters and errors corresponding to each set of target spectra are saved. After the structure parameters are filtered by minimizing the errors, they are imported into the RCWA program to calculate the absorption spectra of the corresponding structures and compared with the target input spectra again for filtering. Finally, a set of structure parameters with good performance and their corresponding absorption spectra are obtained, as shown in Figure 8a.

The blue circles in Figure 8a show the target spectra input to the BNN with an absorptivity of 1 at 1.54 μm, and the four coefficient variables of the Gaussian function used in the range of 3 to 14 μm are 0.07, 0.82, 6.5, and 0.9, respectively. We obtained the inversely designed geometric parameters of *L* = 1.02 μm, *D* = 0.42 μm, *H*_1_ = 0.33 μm, *H*_2_ = 0.0088 μm, and *H*_3_ = 0.65 μm by inputting the Gaussian spectra into the trained BNN. The red squares and the red line represent the actual simulated absorption spectra of the designed structure. The deviations of the positions and values of the absorption peaks are 0.35 μm and 0.03, respectively. Finally, the average absolute error MAE = 0.0718 of the two absorption spectra is obtained by calculation, and the accuracy of the inverse design reaches 92.82%. As shown in Figure 8a, the full absorption spectra of the structure at 1~14 μm are plotted, from which it can be seen that the structure shows three distinct absorption peaks at 1.54 μm, 2.9 μm and 6.5 μm, respectively. The narrower absorption peak (*ε* = 0.88) at 1.54 μm is more consistent with the target effect of single-wavelength laser stealth. Combining the blackbody background radiation at a temperature of *T*, the average emissivity *ε*_avg_ of the material in the wavelength band [*λ*_1_,*λ*_2_] can be calculated by the following equation:(5)$$\varepsilon_{avg}=\int_{\lambda_{1}}^{\lambda_{2}}\varepsilon (\lambda )I_{BB}(\lambda ,T)d\lambda /\int_{\lambda_{1}}^{\lambda_{2}}I_{BB}(\lambda ,T)d\lambda$$
where *I*_BB_ is the blackbody radiation intensity given by Planck’s law. *ε*(*λ*) is the spectral emissivity of the metasurface. The averaged emissivities of the structure in the two atmospheric windows of 3~5 μm and 8~14 μm are calculated to be 0.12 and 0.17, both of which are tailored to be below 0.2 with significant suppression of IR signature, whereas the average emissivity in the non-atmospheric window band 5~8 μm is 0.61, the average emissivity in the main heat dissipation band 6~7 μm reaches 0.81, which improves its ability of radiative heat dissipation, ensuring efficient thermal management performance.

However, all the current calculations are based on the normal incidence of TM waves, and it is impossible to consider only the normal incidence case in real engineering applications, especially the thermal radiation in the IR band needs to be considered at larger angular ranges. In the structure designed based on BNN, multi-angle incidence cases other than normal incidence are also considered, as shown in Figure 8b for the absorptivity of IR radiation from 1 to 14 μm under different angular incidence conditions. It can be seen that the structure always maintains a low absorptivity level of IR radiation in two atmospheric windows (3~5 μm and 8~14 μm) under different angular incidence conditions. At the same time, there is always an absorption peak in the non-atmospheric window (5~8 μm), which proves that the structure also has the performance of multi-band stealth in a wide angular range from −80° to 80°.

### 3.4. Physical Mechanism Analysis

We further consider the underlying physical mechanism for the multi-band IR stealth of the metasurface. Figure 9a–d show the distributions of magnitude of the electric field **|E|** on the plane of *y* = 0 at 1.54 μm, 2.9 μm, 6.5 μm and 14 μm under TM wave normal incidence. At 1.54 μm, **|E|** is mainly distributed between the ultrathin Ag layer and the Al substrate, which indicates the formation of the Fabry–Perot resonance cavity [54] between the Ag and Al substrate. Thus, the high absorption of incident radiation at 1.54 μm is attributed to the Fabry—Perot resonance effect. Numerical results further show that this absorption peak is closely related to the thickness of the Ag layer. At 6.5 μm, **|E|** is mainly distributed in the circular hole, indicating that cavity resonances [55] are formed in the circular hole. Thus, the high emissivity or absorptivity of the metasurface at 6.5 μm is caused by the cavity resonances supported by the hole. At 2.9 μm, higher **|E|** is observed between the ultrathin Ag layer and the Al substrate as well as in the circular hole, which is due to the coupling between the Fabry—Perot resonance and the cavity resonance. This coupling effect explains the high absorption peak at 2.9 μm, as shown in Figure 9c. For those spectral ranges of low emissivity, we especially consider the wavelength of 14 μm in Figure 9d. As shown, the electric field intensity is much smaller at most of the locations, Fabry—Perot like resonances or cavity resonances are not clearly observed, which explains the lower emissivity or absorptivity at 14 μm.

## 4. Conclusions

In this study, a Ge/Ag/Ge multilayer circular-hole metasurface is designed and optimized based on a deep neural network for multi-band compatible IR stealth of military targets. Low emissivities in the two atmospheric windows (3~5 μm and 8~14 μm) and an emissivity peak in the non-atmospheric window (5~8 μm) are achieved. In addition, a high and narrow absorption band is achieved at 1.54 μm for the NIR laser-guided missiles’ detection. The structure optimized by the neural network has a narrow-band absorptivity as high as 0.87 at 1.54 μm. The average emissivity in the two atmospheric windows is controlled at a lower level of 0.11 and 0.16, respectively. The peak emissivity in the non-atmospheric window reaches 0.87, and the average emissivity is above 0.6, ensuring the need for radiative thermal management. Compared with the traditional trial-and-error and parametric scanning methods, the deep neural network as a data-driven method greatly improves design efficiency and saves time and computational resource costs. The above spectral properties also apply to large incident/emission angles. The underlying mechanisms for the multi-band compatible radiative properties are the Fabry—Perot resonances effect and the cavity resonances excited in the metasurface. Our work also demonstrates that DL is efficient for the design and optimization of metasurfaces for multi-band stealth.

## Figures and Tables

**Figure 1 nanomaterials-13-01030-f001:**
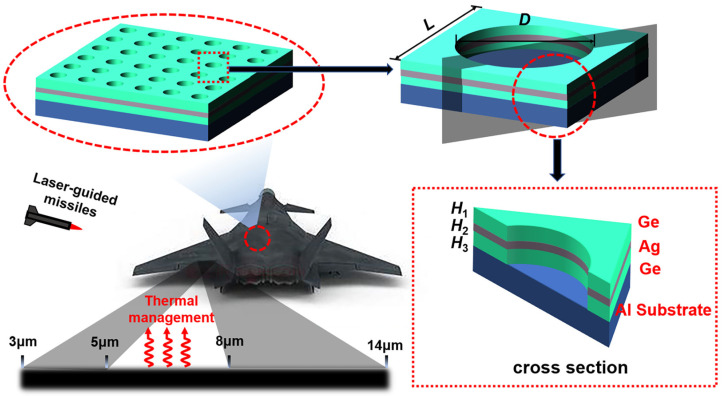
Schematic diagram of the Ge/Ag/Ge multilayer circular-hole metasurface for multi-band IR stealth and thermal management. Thin films of Ge, Ag and Ge with periodic circular holes are stacked on an Al substrate, *L* is the periodicity, *D* is the diameter of the hole, *H*_1_, *H*_2_ and *H*_3_ are the thickness of the top Ge layer, the middle Ag layer and the bottom Ge layer respectively. Assisted with DL, the metasurface is optimized to achieve low emissivities in the two atmospheric windows, high emissivities between the two atmospheric windows for thermal management, and a high absorptivity at near-infrared wavelength for laser guidance.

**Figure 2 nanomaterials-13-01030-f002:**
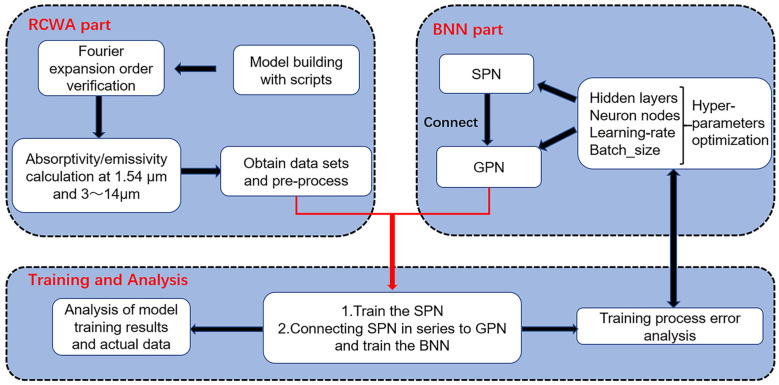
Schematic diagram of the entire workflow.

**Figure 3 nanomaterials-13-01030-f003:**
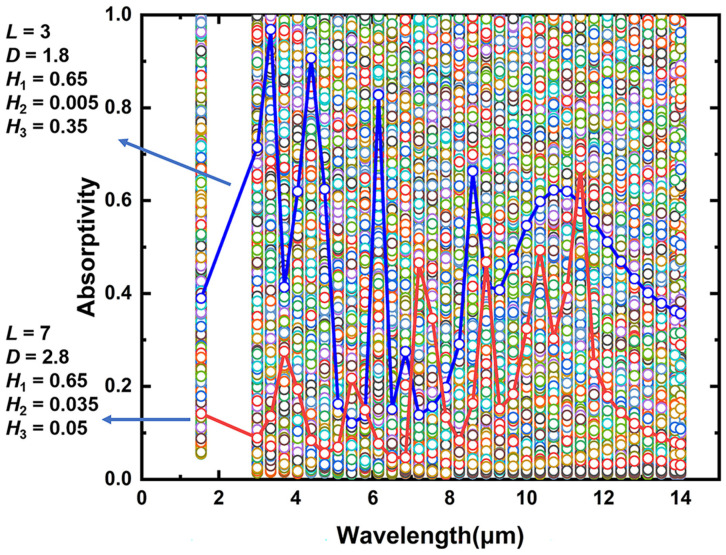
Distribution of 9800 absorption spectra datasets, the absorption spectra of the metasurface with two randomly chosen sets of parameters are shown.

**Figure 4 nanomaterials-13-01030-f004:**
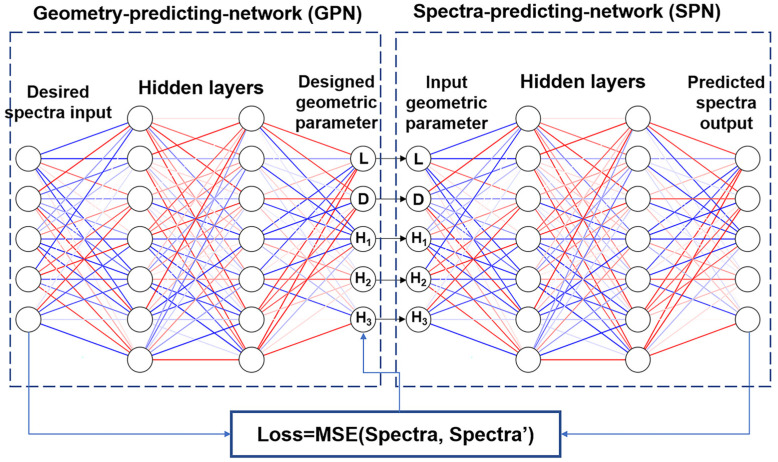
Simplified schematic of a multilayered bidirectional neural network (BNN) model consisting of a GPN connected by a pre-trained SPN.

**Figure 5 nanomaterials-13-01030-f005:**
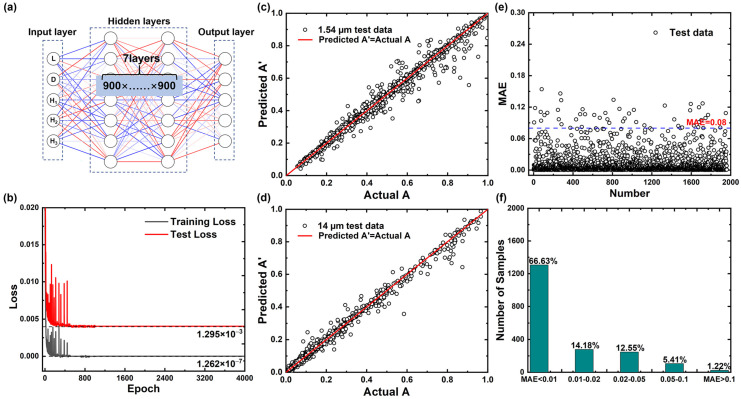
(**a**) Framework of the SPN; (**b**) the training and test loss curves; (**c**,**d**) scatter plot of the actual absorptivity and predicted absorptivity for the 1960 groups of test data at the wavelengths of 1.54 µm and 14 µm, respectively; (**e**) MAE between the actual absorptivity and the predicted absorptivity of the metasurface; (**f**) statistical distribution of test-set samples in different MAE ranges.

**Figure 6 nanomaterials-13-01030-f006:**
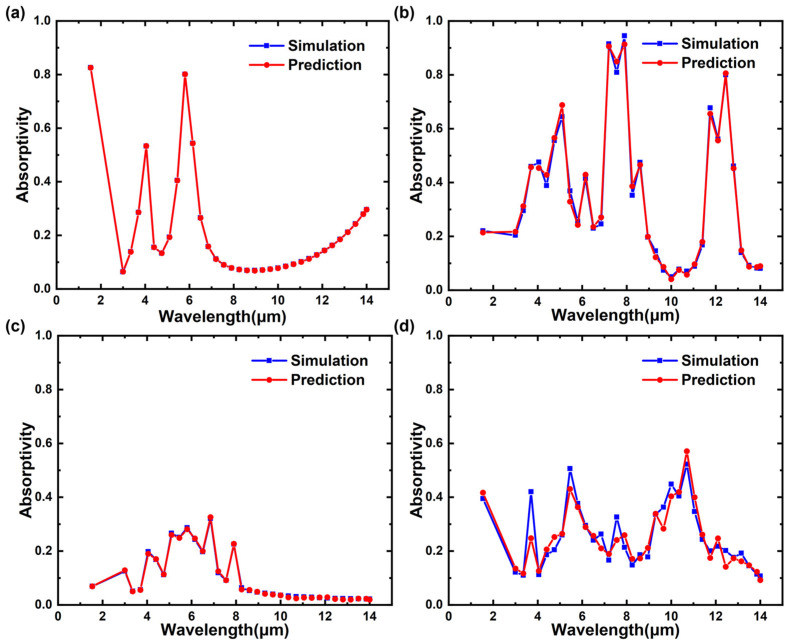
Comparison of the simulated and predicted spectral absorptivity of four randomly selected test-set samples at the target wavelengths. The geometric parameters of the four structures are as follows: (**a**) *L* = 9, *D* = 0, *H*_1_ = 0.95, *H*_2_ = 0.005, *H*_3_ = 0.35; (**b**) *L* = 7, *D* = 5.6, *H*_1_ = 1.55, *H*_2_ = 0.02, *H*_3_ = 1.25; (**c**) *L* = 7, *D* = 2.8, *H*_1_ = 0.35, *H*_2_ = 0.04, *H*_3_ = 0.05; (**d**) *L* = 9, *D* = 7.2, *H*_1_ = 0.05, *H*_2_ = 0.01, *H*_3_ = 1.25. The red dotted line denotes the predicted value of the metasurface, while the blue dotted line plot denotes the actual value obtained by the RCWA.

**Figure 7 nanomaterials-13-01030-f007:**
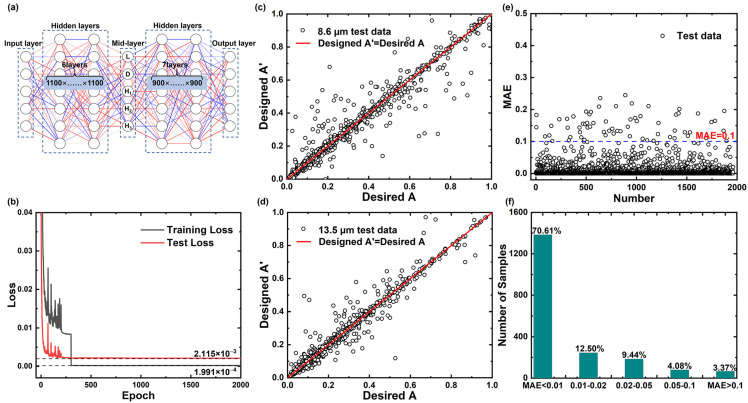
(**a**) Framework of the GPN; (**b**) the training and test loss curves; (**c**,**d**) scatter plot of the designed absorptivity and desired/targeted absorptivity for the 1960 groups of test data at wavelengths of 8.6 µm and 13.5 µm; (**e**) MAE between the designed absorptivity and desired/targeted absorptivity; (**f**) statistical distribution of test-set samples in different MAE ranges.

**Figure 8 nanomaterials-13-01030-f008:**
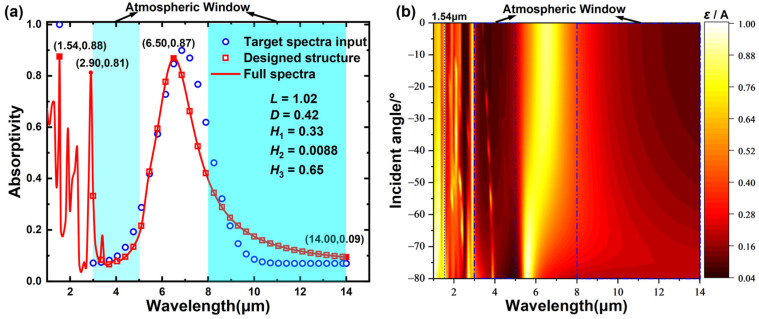
(**a**) IR absorption spectra of the optimized structure, the scattered dots denote the absorptivity of the designed structure (red squares), and the target spectra (blue circles), the geometric parameters of the designed structure are also included; (**b**) contour plot of the absorptivity as a function of wavelength and incident angle.

**Figure 9 nanomaterials-13-01030-f009:**
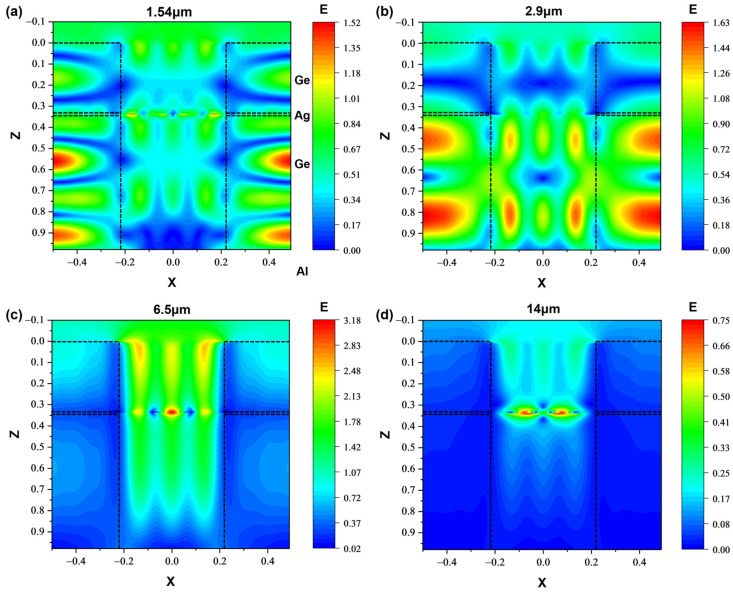
Distribution of magnitude of electric field **|E|** in the plane of *y* = 0 for the structure at wavelengths of (**a**) 1.54 μm; (**b**) 2.9 μm; (**c**) 6.5 μm and (**d**) 14 μm.

## Data Availability

The data presented in this work are available from the corresponding author.

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
