# Peer review of "Deep Learning Assisted Optimization of Metasurface for Multi-Band Compatible Infrared Stealth and Radiative Thermal Management"

_nanomaterials, 2023, doi:10.3390/nano13061030_

Round 1

Reviewer 1 Report

This manuscript deals with a deep-learning approach to designing metasurfaces for radiative thermal management. The authors used RCWA and BNN for obtaining designs that have improved performance in terms of emissivity spectra. The calculation results have interesting performances. I have a few comments on improving the quality of this manuscript.

- The authors used Ge/metal/Ge layers with micropatterns. However, there are simpler versions (e.g., simple Ge/metal without micropatterns (Opt. Express 29, 31364 (2021), and many other thinfilm stacks, etc.) for IR emission. The authors should add those references and comment on that. 

- In the case of infrared stealth and IR radiation, since fine-tuning such emissivity is hard to implement, some researchers are trying to use polarization. The reviewer is wondering if the approach the authors did also consider such a kind of polarization effect. The authors could comment on that in the introduction or conclusion section.

- Why did you use BNN instead of conventional ANN or CNN/GAN? The detailed explanation could be helpful for readers.

- Some figures are very messy to understand. Please simplify it. 

Reviewer 2 Report

Topic of the paper are Ge/Ag/Ge multilayered structures, featuring a hole with a diameter in the micrometers range. Such structures are numerically simulated, demonstrating appealing properties in view of their application as stealth metasurfaces (e.g., low reflectivity in the IR atmospheric window ranges accompanied with strong absorption at 1.5 um and reasonable IR emissivity outside the atmospheric windows).

Numerical simulation of stealth metasurfaces is certainly a non original subject. Thanks to the wide diffusion of powerful numerical methods, including the RCWA used by the Authors, a large number of papers has appeared in the last years, where more or less complex, and more or less practically feasible, architectures and materials have been proposed and basically demonstrated.

Added value of the present paper is the implementation of an optimization method based on deep learning. Despite the limited scope of the optimization (only geometrical parameters are considered, materials and general architecture of the metasurface are kept fixed), the discussed application of deep learning approaches to the field of metasurfaces can be of interest and makes the manuscript worth to be considered for publication. Moreover, presentation is clear and conclusions are rather well supported by results.

However, prior to acceptance, Authors must consider the following points.

1.     The metasurface sketch presented in Fig. 1 is unclear and must be replaced with a more readable scheme, with plan and cross section views. I suggest Authors to remove the panel intended to explain the main system properties, which, by the way, is not well discussed nor mentioned in the text.

2.     Authors must explain in the text how they assessed “accuracy of the calculation” mentioned at line 119.

3.     I found very difficult to understand the dataset representation given in Fig. 3. Authors must clarify either in the caption or in the text the meaning of the colored dots shown in the figure.

4.     Sentence at lines 141-143 sounds unclear to me and should be revised.

5.     The MAE acronym introduced at line 177 is clarified only at line 184: Authors should first define and then use the quantity.

6.     I could not understand the term “tandem architecture” at line 212: Authors should explain it in the text. 

7.     Architecture of the approach shown as an inset in Fig. 7a is definitely too small and cannot be understood: Authors must enlarge the inset, or at least duly increase the lettering size.

8.     The “not large” deviation of peak position and size mentioned at line 287 should be quantified.

9.     A few typos have been detected, e.g. “combing” instead of “combining” (two instances). Authors are invited to carefully check for style, language, and typos.

Round 2

Reviewer 1 Report

The revised version is well-organized based on all the reviewer's concerns.